# Antibacterial Activity and Mechanisms of Essential Oil from *Citrus medica L. var. sarcodactylis*

**DOI:** 10.3390/molecules24081577

**Published:** 2019-04-22

**Authors:** Ze-Hua Li, Ming Cai, Yuan-Shuai Liu, Pei-Long Sun, Shao-Lei Luo

**Affiliations:** 1College of Biotechnology and Bioengineering, Zhejiang University of Technology, Hangzhou 310014, China; 1111405109@zjut.edu.cn; 2Department of Food Science and Technology, Zhejiang University of Technology, Hangzhou 310014, China; 2111626030@zjut.edu.cn; 3Department of Chemical and Biological Engineering, Hong Kong University of Science and Technology, Hong Kong; bxxylzh@163.com

**Keywords:** finger citron, essential oil, antibacterial, mechanism

## Abstract

In this work, antibacterial activity of finger citron essential oil (FCEO, *Citrus medica L. var. sarcodactylis*) and its mechanism against food-borne bacteria were evaluated. A total of 28 components in the oil were identified by gas chromatography-mass spectrometry, in which limonene (45.36%), γ-terpinene (21.23%), and dodecanoic acid (7.52%) were three main components. For in vitro antibacterial tests, FCEO exhibited moderately antibacterial activity against common food-borne bacteria: *Escherichia coli*, *Staphylococcus aureus*, *Bacillus subtilis* and *Micrococcus luteus*. It showed a better bactericidal effect on Gram-positive bacteria than Gram-negative. Mechanisms of the antibacterial action were investigated by observing changes of bacteria morphology according to scanning electron microscopy, time-kill analysis, and permeability of cell and membrane integrity. Morphology of tested bacteria was changed and damaged more seriously with increased concentration and exposure time of FCEO. FCEO showed a significant reduction effect on the growth rate of surviving bacteria and lead to lysis of the cell wall, intracellular ingredient leakage, and consequently, cell death.

## 1. Introduction

Food spoilage caused by microorganisms is always a main public health issue in our daily life. It leads to shelf life reduction, foodborne diseases, and economic loss in the food industry. For example, *Staphylococcus aureus* (*S. aureus*) is mainly responsible for food poisoning, toxic shock syndrome, endocarditis, and osteomyelitis [1]. In decades, synthetic additives have been widely used. However, the addition of synthetic preservatives is more restricted because of their toxic effects. Therefore, a strong interest in using natural substances for food preservation as an alternative has appeared. In recent years, essential oils have attracted lots of scientific interests because they exhibit a wide spectrum of bioactivities, such as antibacterial, antifungal, antiviral, antioxidant, and insecticidal activities [2,3].

Finger citron essential oil (FCEO) is a well-known oil with pleasant odor and flavor intensity, obtained from the epicarp and mesocarp of the fresh fruit of *Citrus medica L. var. sarcodactylis*. Nowadays, FCEO is in great demand in pharmaceutical, food, perfumery, and cosmetic industries. It has proven its analgesic, anxiolytic, neuroprotective effects and antimicrobial efficacy against bacteria and molds [4,5]. However, to the best of our knowledge, few have reported on the action mechanisms of FCEO on the growth inhibition of microorganisms.

The aim of this study is to demonstrate the compositions of FCEO, evaluate its antibacterial activity, and explore its mechanisms of antibacterial action against two respective food-borne microorganisms by scanning electron microscope (SEM), time-kill analysis, and cell membrane permeability, as well as the integrity of the cell membrane.

## 2. Results and Discussion

### 2.1. Chemical Characterization of FCEO

28 compounds were identified by gas chromatography-mass spectrometry (GC-MS), including 16 monoterpenoid hydrocarbons, 8 sesquiterpenoid hydrocarbons, 3 acids, and 1 ester, as shown in Table 1. Main components of the oil were limonene (45.36%), γ-terpinene (21.23%), and dodecanoic acid (7.52%). Limonene widely exists in lemon and other citrus fruits. Linalool, 0.47%, in the oil, is the main component for characteristic aroma of citrus, with a pleasant odor and mainly separated from scented herbs, rosewood, and citrus. Moreover, linalool is a chemical intermediate to yield vitamin E, hypotensive, perfumed product and cleaning agent. Most studies indicate that FCEO contains the major components limonene (33.8%~57.7%) and γ-terpinene (22.4%~33.7%), along with 3-carene, α-pinene, β-pinene, α-thujone, and terpinolene [6]. Russo et al. [7] reported FCEO comprised monoterpene and sesquiterpene hydrocarbons (limonene, γ-terpinene) and oxygenated derivatives (linalool, linalyl acetate). However, the ratio of these components in the oil compared with the published data were different. Various factors could affect the concentrations, such as harvest time, geographic origin, and agro-climatic conditions of the regions.

### 2.2. Antibacterial Activity of FCEO

#### 2.2.1. Effects of FCEO on Bacteria

According to previously published studies, diameters of inhibition zone (DIZ) were appreciated as follows: Not sensitive (diameter ≤ 8.0 mm), moderately sensitive (8.0 < diameter < 14.0 mm), sensitive (14.0 < diameter < 20.0 mm), and extremely sensitive (diameter ≥ 20.0 mm) [8]. The results showed the essential oil had certain antibacterial activity on all of the tested pathogens, with DIZ of the maximum value for *S. aureus* (19.2 ± 2.1 mm), followed by *Bacillus subtilis* (*B. subtilis*, 16.3 ± 1.3 mm), *Micrococcus luteus* (*M. luteus*, 16.1 ± 0.4 mm) and *Escherichia coli* (*E. coli*, 11.2 ± 0.9 mm), as shown in Table 2. On the contrary, limonene exhibited moderate antibacterial activity. Van Vuuren and Viljoen [9] found that the antibacterial effect of limonene could be enhanced in the essential oil, compared with being used alone. It indicated that a synergistic effect might occur in the oil and thus potentiate its biological activity. Indeed, an advantageous synergistic effect of the essential oils constituents has often been observed [8]. Moreover, the oil showed better activity against Gram-positive bacteria than Gram-negative bacteria. It might be attributed to the structure of the bacterial membrane that Gram-negative bacteria possess an outer membrane with the presence of lipopolysaccharide molecules, which provide a hydrophilic surface [10]. The surface acts as a penetration barrier that blocks macromolecules and hydrophobic compounds penetrate into the target cell membrane [11]. Accordingly, Gram-negative bacteria are relatively resistant to hydrophobic antibiotics.

#### 2.2.2. Minimal Inhibitory Concentration (MIC) and Minimum Bactericidal Concentration (MBC)

FCEO is a bacterial inhibitor and bactericide against four tested bacteria. MICs and MBCs were in the range of 0.625~2.5 mg/mL and 1.25~2.5 mg/mL, respectively, as shown in Table 2. FCEO exhibited the best bactericidal activity against *S. aureus*, with both minimum MIC and MBC. Gram-positive bacteria showed more sensitive than Gram-negative bacteria, which was in accordance with the results of DIZ. These results were similar with other researchers that citrus essential oil inhibited *S. aureus*, *B. subtilis,* and *E. coli* with MICs of 1%, 2% and 2% (*v*/*v*), respectively [12]. In contrast, Ghabraie et al. [13] tested bergamot essential oil against *E. coli* and *S. aureus* with MIC of 1.25 and 2.5 mg/mL, respectively. The antibacterial activity of FCEO might attribute to monoterpene, alcohol, phenols, and other minor compounds. It has been reported that limonene possessed antibacterial, antiviral, antifungal and antioxidant activities, as mentioned in 2.2.1, while linalool and linalyl acetate exhibited antibacterial and antiviral activities [2]. Minor components possibly produce a synergistic effect to inhibit microorganisms as well. These results indicate FCEO is a potential bacterial inhibitor with a broad antibacterial spectrum, but the mechanisms of the antibacterial action need further study. 

### 2.3. Antibacterial Mechanisms of FCEO Against E. coli and S. aureus 

#### 2.3.1. The Effect of FCEO on Morphological Change

Bacteria were treated with FCEO at concentrations of MIC and 2×MIC, respectively; untreated bacteria were set as control. Morphological changes of both treated and untreated bacteria were investigated by SEM. The results were shown in Figure 1. Untreated *E. coli*, as shown in Figure 1A, exhibited distinctive features characterized by regular rod-shaped, intact surface and striated cell walls. In contrast, most of the treated bacteria became irregular and shriveled of different degree, as shown in Figure 1B,C. Moreover, cell walls of the tested bacteria treated at 2×MIC level showed more severe morphological destruction than those at a concentration of MIC. Untreated *S. aureus* cells, as shown in Figure 1D, were spherical, regular and intact and have a smooth surface. When exposed to FCEO for 4 h, the cell membranes were pitted and shriveled, with holes on the surface, as shown in Figure 1E,F. In addition, bacterial aggregation could be observed. The changes of tested bacteria were due to the effect of FCEO, which could cause the destruction of the cell membrane of *E. coli* and *S. aureus* and the losses of intracellular materials. Microbial organisms were killed probably because the cytoplasmic membrane was disrupted or permeated through an interfacial contacting inhibitory effect that occurred on the surface of microspheres [11]. Both tested bacteria showed the essential oil-induced deformation of target cells occurred in a dose-dependent manner, which was also supported by other studies [14]. 

#### 2.3.2. Effect of FCEO on the Viability

In order to evaluate the inactivation kinetics of FCEO, time-kill assays were performed, expressed as a logarithm of viable counts in Figure 2. Untreated *E. coli* increased from 5.1 to 8.3×Log_10_ CFU/mL and transited into stationary phase after 6 h. Treated bacteria decreased sharply in the first 4 h and maintained steadily at about 2.5×Log_10_ CFU/mL. The inhibition rate of *E. coli* reached 99.7% with the existence of essential oil at the concentration of MIC. The curve of tested *E. coli* at 2×MIC level was similar to that at MIC. Untreated *S. aureus* increased from 5.5 to 8.3×Log_10_ CFU/mL in the cultivation time of 8 h. Afterward, the number of viable cells kept stable and slowly decreased to 7.8×Log_10_ CFU/mL after 24 h. Compared with the control, treated *S. aureus* decreased significantly. In the first 2 h, the numbers of viable cells of *S. aureus* treated at MIC and 2×MIC both decreased to approximately 2.4×Log_10_ CFU/mL and maintained stable. The results showed FCEO had a fast killing effect on growth of *S. aureus*, with a bactericidal effect after 2 h of incubation, whereas 4 h for *E. coli* to achieve a lethal effect. This result was in accordance with the results of SEM that *S. aureus* was more sensitive than *E. coli*.

#### 2.3.3. The Effect of FCEO on Cell Membrane Permeability

Cell membrane permeability was determined as the relative electric conductivity. As shown in Figure 3A, the relative electric conductivity of the control slightly increased. It might be because of normal lysis and death of the bacteria. On the contrary, the electric conductivity of tested *E. coli* increased rapidly in the first hours. The growth speed trended to slow down after about 11 h. At the end of the assay, the relative conductivity of bacteria at concentrations of MIC and 2×MIC reached 37.04 ± 3.60% and 46.05 ± 2.64%, respectively, compared with 4.94 ± 0.58% of control. It showed relative electric conductivity of tested bacteria increased with the oil concentration and treatment time increasing. Similar trends were observed for *S. aureus*, but it showed higher conductivity than that of *E. coli*. After exposed to oil for 12 h, the relative conductivity of *S. aureus* at the control, MIC, and 2×MIC were 10.01 ± 1.66%, 63.98 ± 3.00%, and 80.59 ± 3.65%, respectively. It showed leakage of electrolytes occurred because of disruption of cell permeability caused by FCEO. Cells depend on cytoplasmic membrane to block small ions and keep normal metabolism, including solute transport, management of turgor pressure and motility [15]. Hence, even minor variations to the structure of the membrane can dramatically affect cell metabolism and result in death [16]. Essential oil can increase the permeability of bacteria membrane, leading to leakage of the intracellular ingredient. According to previous studies, phenolic existing in the FCEO can disrupt the cell membrane, interfere with cellular energy (ATP) generation system, and disrupt the proton motive force, eventually cause leakage of internal contents of the cell [17,18]. 

#### 2.3.4. The Effect of FCEO on Integrity of the Cell Membrane

Proteins and nucleic acids are extremely important macromolecules for cells referring to cellular structure and genetic information [19]. Determination of absorbance at 260 nm for nucleic acid and proteins is an indicator of membrane integrity [20]. Figure 4 shows the effects of essential oil on the integrity of cell membrane at MIC and 2×MIC concentrations for 7 h, respectively. The absorbance values of *E. coli* increased significantly from 0.177 ± 0.025 to 0.817 ± 0.032 at MIC and 0.280 ± 0.017 to 0.920 ± 0.026 at 2×MIC, while the control increased from 0.043 ± 0.010 to 0.351 ± 0.010. Release of cell constituents increased significantly with the time and increased concentrations of FCEO. For *S. aureus*, during the period of treatment, absorbance values were in the range of 0.067 ± 0.008~0.338 ± 0.021, 0.237 ± 0.021~0.770 ± 0.046 and 0.267 ± 0.015~0.900 ± 0.036 for oil concentration at the control, MIC, and 2×MIC, respectively. This result indicates FCEO can affect the integrity of the membrane, leading to the leakage of nucleic acids and proteins through the membrane, and consequently, cell death. Another probability is that essential oil penetrates through cytoplasmic membrane and especially damages mitochondrial membranes, and after that, the mitochondria produce free radicals, which oxidize and damage lipids, proteins and DNA [21].

## 3. Materials and Methods

### 3.1. Materials

Fresh finger citrons were purchased from Jisobo Biological Technology Co., Ltd, Jinhua, Zhejiang Province, China. The fruits were ground and hydro-distillated by using a Clevenger-type apparatus for 3 h. The distillate was dried with anhydrous sodium sulfate overnight and stored at 4 °C in dark for further use. The essential oil obtained from finger citron was light yellow with a yield of 0.9%.

### 3.2. Chemicals and Microorganisms

Ciprofloxacin, standard substance of limonene, nutrient agar (NA), nutrient broth (NB) and Mueller Hinton agar (MHA) medium were purchased from Aladdin Industrial Corporation (Shanghai, China). Dimethyl sulfoxide (DMSO), glutaraldehyde, tertiary butyl alcohol, and glucose were from Macklin Biochemical Co., Ltd (Shanghai, China). Anhydrous sodium sulfate, sodium chloride, sodium dihydrogen phosphate (NaH_2_PO_4_·2H_2_O), disodium hydrogen phosphate (Na_2_HPO_4_·12H_2_O), diethyl ether, methanol and ethanol used were of analytical grade. 

Three Gram-positive bacterial strains (*Bacillus subtilis* ATCC 6633, *Staphylococcus aureus* ATCC 6538 and *Micrococcus luteus* ATCC 4698) and 1 Gram-negative bacterial strain (*Escherichia coli* ATCC 25922) were obtained from China Center of Industrial Culture Collection (CICC). Bacterial strains were cultured at 37 °C on MHA.

### 3.3. GC-MS Analysis

Analysis of essential oil was carried out using GC-MS (Trace 1300 Gas Chromatograph equipped with an ISQ LT Single Quadrupole Mass Spectrometer Thermo Fisher Scientific, Waltham, MA USA). GC-MS analysis was performed in the electron impact ionization mode (70 eV) with *m*/*z* range 50~500 u. Temperatures of the injector and ion source were set at 250 and 280 °C, respectively. TG-5 MS AMINE GC Column (30 m × 0.25 mm × 0.25 μm) was used. The oven temperature was programmed as follows: Initial oven temperature was 50 °C, holding for 1 min, then raised to 130 °C at 5 °C/min, holding for 0.5 min, then to 250 °C at 15 °C/min, holding for 10 min. Carrier gas was helium at a constant flow rate of 1.0 mL/min. The essential oil was diluted in ethanol of 1:500 before analysis and was injected 1.0 μL into gas chromatograph with a split ratio of 10:1. EO constituents were identified by retention indices (RI) and NIST 2.0 version of the library [22]. 

### 3.4. Evaluation of Antibacterial Activity of FCEO

#### 3.4.1. Antibacterial Activity of FCEO

Antibacterial activity of essential oil and its main component limonene were tested using agar diffusion method with some modifications [23]. Briefly, 100 μL bacterial suspension (approximately 1 × 10^7^ CFU/mL) was spread on NA medium. A sterile filter paper disc (diameter = 6 mm) containing 5 μL of the sample was placed on the surface of plate. Incubating at 37 °C for 24 h, all the plates were observed for zones of growth inhibition and the diameters in millimeters of these zones were measured. Ciprofloxacin was used in parallel experiments as a positive control.

#### 3.4.2. Determination of MIC and MBC

MIC was determined by the method of Silveira et al. [24] with a few modifications. Briefly, the stock of sample was diluted in 5% (*v*/*v*) DMSO. No detrimental effect on bacterial growth was observed at this concentration. Two-fold serial dilutions of samples were prepared in sterile NB ranging from 0.039–10 mg/mL. Afterward, 180 μL solution was mixed with 20 μL bacterial suspension (approximately 10^7^ CFU/mL) in the 96-well plates. A negative control test containing inoculated broth only supplemented with DMSO and a positive control containing Ciprofloxacin were also performed. The plates were incubated at 37 °C for 24 h. MBC was measured by subculture of 50 μL from each well with no visible bacterial growth on NA plates after incubating at 37 °C for 24 h. 

### 3.5. Demonstration of Antibacterial Mechanisms of FCEO 

#### 3.5.1. SEM Analysis of Morphological Changes 

SEM was used to observe the morphological changes according to the method described by Bajpai, Sharma, and Baek [20] with a few modifications. The bacteria were incubated in NB medium at 37 °C for 8 h and centrifuged 5000 r/min. The cells were resuspended with phosphatic buffer solution (PBS, 0.1 M, pH 7.4) of approximately 10^8^ CFU/mL. The suspension mixed with essential oil at different concentrations of the control, MIC, and 2×MIC. DMSO was added to 5% (*v*/*v*) of the final volume, followed by incubating at 37 °C for 4 h. The suspensions were centrifuged (8000 r/min, 5 min) and washed by 0.1 M PBS twice. Afterward, the cells were transferred into 2.5% (*v*/*v*) glutaraldehyde in 0.1 M PBS at 4 °C for another 4 h. The cells were washed by 0.1 M PBS three times and dehydrated in sequential graded ethanol concentrations of 30%, 50%, 80%, 90%, and 100%. Finally, the ethanol was replaced by 100% tertiary butyl alcohol. Samples were sputter-coated with gold in an ion coater for 2 min, followed by microscopic examinations on a SEM (Tescan Vega 3 SBH, Brno, Czech).

#### 3.5.2. Time-Kill Analysis

To investigate the bactericidal effects of FCEO, time-kill analysis was curved as Joray et al. [25] described. After incubating in NB medium at 37 °C for 8 h, bacteria were centrifuged and resuspended in sterile saline to approximately 10^5^ CFU/mL. The bacterial suspension was mixed with NB medium containing 5% DMSO in the presence of FCEO at different concentrations of control, MIC and 2×MIC, respectively. The inoculums were cultured at 37 °C with shake. Samples were taken from the culture at selected time intervals, diluted in sterile saline and cultured on NA medium. Colony-forming units were counted after incubating for 24 h at 37 °C. 

#### 3.5.3. Cell Membrane Permeability

Cell membrane permeability is expressed as the relative electric conductivity. It was measured as Kong et al. [11] described with minor modifications. Bacterial strains were incubated at 37 °C for 8 h, followed by centrifugation at 5000 r/min for 10 min. The cells were washed with 5% glucose solution (containing 5% DMSO, *v*/*v*) and centrifuged until the electric conductivities of the cells were near to that of 5% glucose. The isotonic bacteria were approximately 10^6^ CFU/mL. Conductivities of essential oil added with 5% glucose at different concentrations were measured and recorded as L_1_. The essential oil at different concentrations completely mixed with bacterial suspension and incubated at 37 °C for 12 h. The conductivities were measured per hour and recorded as L_2_. The conductivity of bacteria in 5% glucose treated in boiling water for 5 min was marked as L_0_. The cell membrane permeability is calculated as the following formula:(1)$$\mathrm{Relative}\;\mathrm{electric}\;\mathrm{conductivity}\;\left(\%\right)=\frac{\left(L_{2}-L_{1}\right)}{L_{0}}\times 100\%$$

#### 3.5.4. The Integrity of Cell Membrane

Release of cell constituents into the supernatant was carried out according to the method of Du et al. [26] with a few modifications. Tested microorganisms from 100 mL culture medium were collected by centrifugation for 10 min at 3500× *g*, washed three times and resuspended in 0.1 M PBS, pH 7.4. The cells were added into 100 mL NB medium containing 5% DMSO and essential oil at three different concentrations of control, MIC and 2×MIC. The bacterial suspension was incubated at 37 °C under agitation. After treatment for 0, 1, 2, 3, 4, 5, 6 and 7 hours, cells were centrifuged at 3500× *g* and the supernatant was determined at 260 nm by a UV/Vis Spectrophotometer (TU-1900, Beijing, China), respectively. Correction was made for absorption of the suspension with the same PBS containing the same concentration of the oil after 2 min of contact with tested strains. The untreated cells were corrected with PBS only.

### 3.6. Statistical Analysis

All the assays were carried out triplicate. The data were recorded as mean ± standard. A general analysis of variance (ANOVA) and Duncan’s multiple range tests were performed. Values of *p* < 0.05 were considered to be statistically different.

## 4. Conclusions

To broaden the application of FCEO as a natural antimicrobial agent in the food industry, the study determined its antimicrobial activity and gave insight into its mode of action on *E. coli* and *S. aureus* as representatives from Gram-negative and Gram-positive bacteria. It demonstrates 28 compounds in essential oil which are abundant with limonene, γ-terpinene, and dodecanoic acid. During in vitro antibacterial activities tests, Gram-positive organisms seemed to be more susceptible to the oil than Gram-negative organisms, which is attributed to the structure of the cellular membrane. FCEO induces alterations in the morphology of *E. coli* and *S. aureus*. Permeability and integrity of the cell membrane tests suggest intracellular materials, including small ions, nucleic acids, and proteins, leaked when treated with FCEO at MIC and 2×MIC levels, in a dose-dependent manner.

## Figures and Tables

**Figure 1 molecules-24-01577-f001:**
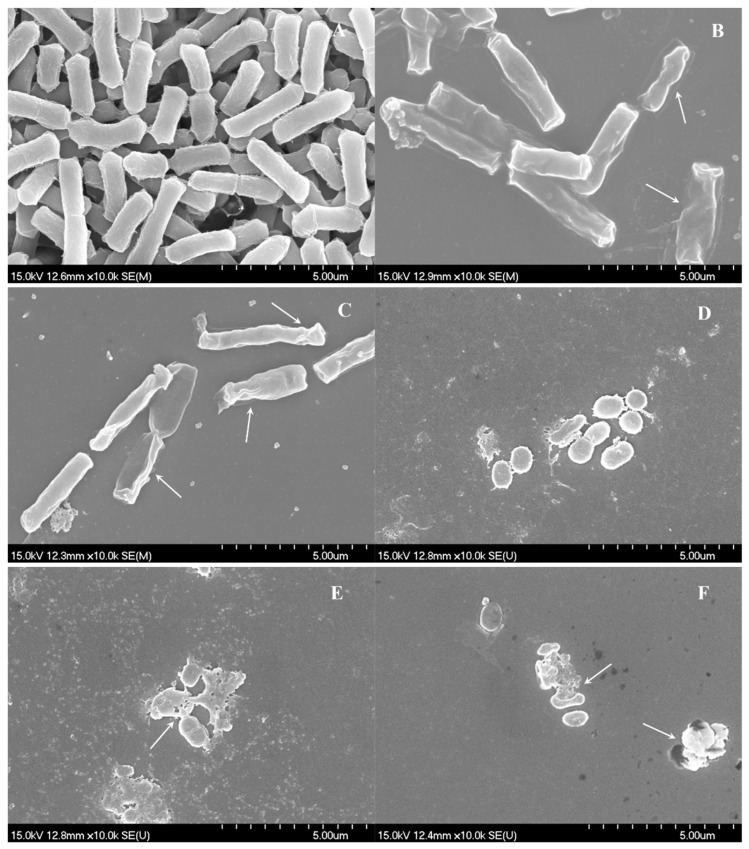
Effects of FCEO on morphological changes of *E. coli* and *S. aureus*. (**A**): Untreated *E. coli*; (**B**): *E. coli* treated with FCEO at minimal inhibitory concentration (MIC); (**C**): *E. coli* treated with FCEO at 2×MIC; (**D**): Untreated *S. aureus*; (**E**): *S. aureus* treated with FCEO at MIC; (**F**): *S. aureus* treated with FCEO at 2×MIC. Arrows show the shriveled appearance and holes on the cell surface.

**Figure 2 molecules-24-01577-f002:**
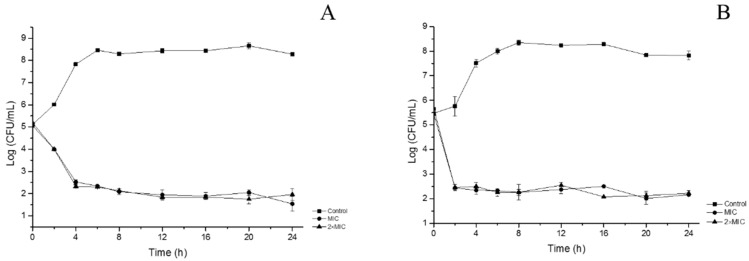
Time-kill analysis of *E. coli* and *S. aureus*. (**A**): *E. coli*; (**B**): *S. aureus*.

**Figure 3 molecules-24-01577-f003:**
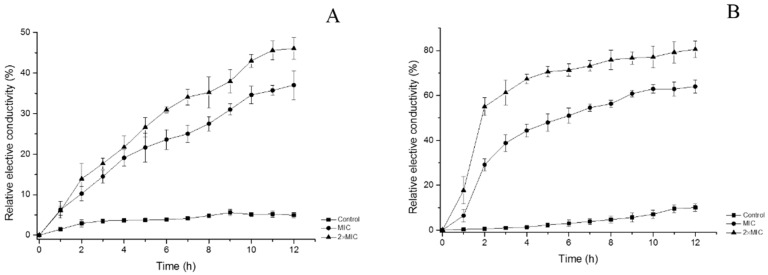
Effects of FCEO on the permeability of *E. coli* and *S. aureus*. (**A**): *E. coli*; (**B**): *S. aureus*.

**Figure 4 molecules-24-01577-f004:**
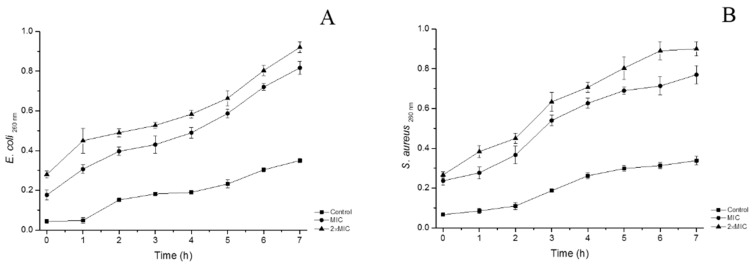
Effects of FCEO on the integrity of cell membrane of *E. coli* and *S. aureus*. (**A**): *E. coli*; (**B**): *S. aureus*.

**Table 1 molecules-24-01577-t001:** Components of the essential oil from finger citron.

No.	Compound	RI ^a^	RI (lab) ^b^	Area%	CAS No.
1	α-Pinene	944	948	1.42	80-56-8
2	3-Carene	951	950	0.69	13466-78-9
3	β-Pinene	953	956	1.15	127-91-3
4	β-Myrcene	980	979	0.81	123-35-3
5	α-phellandrene	983	983	0.63	99-83-2
6	Allo-ocimene	998	993	0.24	673-84-7
7	α-Terpinene	1011	1008	0.56	99-86-5
8	Limonene	1021	1020	45.36	138-86-3
9	cis-β-Ocimene	1024	1024	0.38	3338-55-4
10	γ-Terpinene	1045	1047	21.23	99-85-4
11	Linalool	1080	1081	0.47	78-70-6
12	Terpinen-4-ol	1134	1137	2.35	562-74-3
13	α-Terpineol	1168	1172	2.51	98-55-5
14	Geranial	1173	1174	0.05	141-27-5
15	Carveol	1203	1206	0.35	99-48-9
16	Neral	1212	1214	0.04	106-26-3
17	Geranyl acetate	1353	1352	0.15	105-87-3
18	α-Bergamotene	1427	1427	1.42	17699-05-7
19	δ-Cadinene	1439	1440	0.35	483-76-1
20	Germacrene D	1480	1480	0.7	23986-74-5
21	Caryophyllene	1500	1494	1.6	87-44-5
22	γ-Muurolene	1496	1494	0.19	30021-74-0
23	β-Bisabolene	1500	1500	3.23	495-61-4
24	Dodecanoic acid	1571	1570	7.52	143-07-7
25	Humulene	1581	1579	0.3	6753-98-6
26	α-Bisabolol	1686	1683	1.37	515-69-5
27	Tetradecanoic acid	1769	1769	2.85	544-63-8
28	Hexadecanoic acid	1970	1968	1.46	57-10-3
Total				99.38	

a: Data were obtained from experiments. b: Data were obtained from NIST 2.0 version of the library.

**Table 2 molecules-24-01577-t002:** The antibacterial ability of finger citron essential oil (FCEO) against different microorganisms.

Microorganisms	DIZ * (mm)			MIC (mg/mL)			MBC (mg/mL)		
	FCEO	Limonene	Ciprofloxacin	FCEO	Limonene	Ciprofloxacin	FCEO	Limonene	Ciprofloxacin
Gram-positive									
*B. subtilis*	16.3 ± 1.3 ^b^	10.8 ± 1.2 ^ab^	23.7 ± 0.6 ^a^	0.625	1.25	0.001	2.5	1.25	0.001
*S. aureus*	19.2 ± 2.1 ^a^	11.2 ± 1.0 ^a^	25.0 ± 0.3 ^a^	0.625	0.625	0.001	1.25	1.25	0.001
*M. luteus*	16.1 ± 0.4 ^b^	9.3 ± 0.6 ^bc^	21.7 ± 1.4 ^b^	1.25	2.5	0.001	1.25	5	0.001
Gram-negative									
*E. coli*	11.2 ± 0.9 ^c^	8.6 ± 0.6 ^c^	19.3 ± 0.9 ^c^	2.5	1.25	0.001	2.5	1.25	0.002

* Values represent means of three independent replicates ± SD. Different letters within a column indicate statistically significant differences (*p* < 0.05) for DIZ.

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
