# Peer review of "Antibacterial Activity and Mechanisms of Essential Oil from Citrus medica L. var. sarcodactylis"

_molecules, 2019, doi:10.3390/molecules24081577_

Round 1

Reviewer 1 Report

The revised manuscript is accpetable in its present form.

Author Response

Response 1: Thank you very much for your approval.

Reviewer 2 Report

After reading the corrected manuscript I have to say that table 1 looks worse than in the previous version, even the authors provided the retention indices. Based on the description of the experimental part concerning GC-MS (used column an temperature program) it is impossible to detect beta-pinene before alpha-pinene, as well as trans-beta-ocimene before myrcene. The NIST library is based on the published data. SO, if you want to compare your retention indices with those in this library you should pay attention on the GC conditions. I can not agree with the publication of such a wrong data.

Author Response

Thank you for your comments. Sorry for our careless. We rewrote the results after checking the data carefully and compared with published literatures [1,2], as highlighted in the manuscript.

Reference

1. Peng, C. H.; Ker, Y. B.; Weng, C. F.; Peng, C. C.; Huang, C. N.; Lin, L. Y.; Peng R. Y., Insulin secretagogue bioactivity of finger citron fruit (citrus medica L. var sarcodactylis hort, rutaceae). J. Agr. Food Chem. 2009, 57, (19), 8812-9.

2. Guo J.; Gao Z.; Xia J.; Ritenour M. A.; Li G.; Shan Y., Comparative analysis of chemical composition, antimicrobial and antioxidant activity of citrus essential oils from the main cultivated varieties in China. LWT-Food Sci. Technol. 2018, 97, 825-39.

This manuscript is a resubmission of an earlier submission. The following is a list of the peer review reports and author responses from that submission.

Round 1

Reviewer 1 Report

The reviewed paper concerns the analysis of antibacterial activity as well as the mechanisms of the EO obtained from finger citron fruits. Despite the fact that the aim of this paper is interesting, especially the part concerning the mechanism of antibacterial activity, there so many mistakes, wrong, confusing, and lacking information in two other parts concerning chemical analysis of essential oil and antibacterial activity.

I am very doubtful about the proper identification of the essential oil components. First of all, the authors did not calculate the relative retention indices, which are necessary, in addition to the mass spectra, for correct identification. There are many questions to the compounds identified and included in table 1:

- trans-beta-ocimene and beta-ocimene; If the authors identified (E)-beta-ocimene, what means beta-ocimene? Is it Z-isomer? Retention indices are necessary!

- beta-citral and citral; Citral is the mixture of two compounds, neral and geranial. So, it is necessary to specify which isomer you identified. beta-Citral is the synonym of neral, but to be sure whether it is correct the retention indices are necessary!

- beta-linalool; D-germacrene; D-cadinene – all are wrong names!

- gamma-pinene (results and discussion) – there is no such a compound!

Since the diagnostic index for good quality citrus oils largely depends on the quantities of two types of compounds; acyclic monoterpene hydrocarbons and acyclic oxygenated monoterpenoids, so, instead to divide the identified components into alkanes (hydrocarbons would be much better), alcohols, aldehydes….., will be much better to mention about mono- and sesquiterpenoids.

In the abstract and conclusions we can read that ‘FCEO exhibited strong antibacterial activity’. But, when one see the data included in table 2, the activity of this essential oil is really weak! 625 microg/ml is really weak! There is also no information about the positive control for antibacterial activity assay. And more, the activity of the major components, at least limonene, should also be determined.

Are you sure that the FCEO at concentration (MIC and 2xMIC), taken for the studies concerning the mechanism of antibacterial activity, was soluble in the testing conditions? These are quite high concentrations, and the performed tests were necessary buffers and medium which are water solutions.

Reviewer 2 Report

 This study was on the antimicrobial activity of finger citron essential oil. The composition of the essential oil and its antimicrobial mechanism has been reported in this study. Overall, this manuscript is well organized and the experimental data have been properly described. However, there are some more explanation and revisions needed for the clarity of the manuscript. Please refer to the blow comments.

Detailed comments

L 84: This results → These results

L 84-86: MIC values are different. Thus, the data are not supported.

L 92-92: “but the mechanisms of FCEO against tested bacteria need further study” → Poor description. Please rewrite the sentence.

L 96: “bacteria of treated or untreated” → Please rewrite.

L 97: “Controlled E. coli” →What do you mean by that?

L 111-114: Figure 1 D → Please recheck the photo, where it does not show usual shape of the bacteria. The morphology of Staphylococcus aureus seems different. It is not spherical. Please repeat the experiment.

L 116: “evaluate the killing kinetics” → inactivation kinetics?

L 119: “More than 99.7% E. coli inhibited” → Please rewrite.

L 126-127: “needed for 4 h to achieve” → Please rewrite.

L 180: “3 Gram-positive bacteria” → Please do not begin the sentence with Arabic numbers. Three?

L 199: “paper disk (6 mm)” → diameter? Please specify.